# Analysis of Breast Cancer Differences between China and Western Countries Based on Radiogenomics

**DOI:** 10.3390/genes13122416

**Published:** 2022-12-19

**Authors:** Yuanyuan Zhang, Lifeng Yang, Xiong Jiao

**Affiliations:** 1College of Biomedical Engineering, Taiyuan University of Technology, Jinzhong 030600, China; 2College of Information and Computer, Taiyuan University of Technology, Jinzhong 030600, China

**Keywords:** breast cancer, radiogenomics, China, western countries, the difference

## Abstract

Using radiogenomics methods, the differences between tumor imaging data and genetic data in Chinese and Western breast cancer (BC) patients were analyzed, and the correlation between phenotypic data and genetic data was explored. In this paper, we analyzed BC patients’ image characteristics and transcriptome data separately, then correlated the magnetic resonance imaging (MRI) phenotype with the transcriptome data through a computational method to develop a radiogenomics feature. The data was fed into the designed random forest (RF) model, which used the area under the receiver operating curve (AUC) as the evaluation index. Next, we analyzed the hub genes in the differentially expressed genes (DEGs) and obtained seven hub genes, which may cause Chinese and Western BC patients to behave differently in the clinic. We demonstrated that combining relevant genetic data and imaging features could better classify Chinese and Western patients than using genes or imaging characteristics alone. The AUC values of 0.74, 0.81, and 0.95 were obtained separately using the image characteristics, DEGs, and radiogenomics features. We screened SYT4, GABRG2, CHGA, SLC6A17, NEUROG2, COL2A1, and MATN4 and found that these genes were positively or negatively correlated with certain imaging characteristics. In addition, we found that the SLC6A17, NEUROG2, CHGA, and MATN4 genes were associated with clinical features.

## 1. Introduction

According to the GLOBOCAN estimate for 2020, BC has become the leading cause of global cancer incidence worldwide [1]. New BC morbidity and mortality in China are increasing yearly, with 0.42 million BC patients diagnosed in 2020, accounting for about 18.4% of the global BC cases [2,3]. China has the most significant BC deaths, accounting for approximately 17.1% of all cancer deaths [3]. BC has substantial racial differences in diagnosis, prognosis, and survival [4]. The age of vulnerability for BC in China is between 55 and 60 years old, while the average age of onset in many western countries is between 60 and 70 years old [3,5]. In addition, compared with the United States, the proportion of Chinese BC patients with stage I, negative lymph nodes, positive ER rates, and 5 year survival are lower [6,7,8,9], and the mean tumor size at diagnosis is relatively larger [10]. The understanding and treatment of BC are based mainly on Western research and data. However, arising contrasts in BC epidemiology, histopathological, genetic, and biological across different races may have implications for clinical treatment [11,12]. Therefore, studying the differences between Chinese and Western BC will help to better understand BC’s pathogenesis and provide theoretical support for implementing precision medicine for BC patients in China.

As gene sequencing technology develops by leaps and bounds [13], scientists have explored racial differences in BC at the molecular level. For example, there are differences in the prevalence of BRCA1 and BRCA2 mutations between Asian and Western countries. White patients are more likely to have BRCA1 mutations, while the opposite is true for Chinese patients [14,15]. Chinese patients with BC identified BRCA2 mutations, which occur almost twice as frequently as BRCA1 mutations [15,16]. The gene that contributes the most to the risk of BC is BRCA2, compared to BRCA1 in studies of European or African descent [17]. Through the analysis of high-throughput sequencing data, Chinese BC patients are more prone to somatic mutations such as PIK3CA, PIK3R1, AKT3, and PTEN [18].

MRI is a non-invasive procedure for characterizing and diagnosing BC. Extracting these imaging features allows tumor phenotypes to be described quantitatively. It is known that, compared with white women, Asian female breasts have the characteristics of smaller gland size and dense tissue [19,20,21], but these are far from enough to describe the differences in imaging. Radiogenomics is a discipline that integrates tumor characteristics and genomic data [22,23] to achieve complementary advantages through high-throughput extraction of tumor phenotype characteristics [24], capturing tumor heterogeneity, and correlating it with specific gene expression patterns [25,26]. In a study of patients with liver cancer, Segal et al. demonstrated that the features extracted from CT images reflected the changes in gene expression modules [27]. Zhu et al. found that BC tumor size, blurred margins, and morphological irregularity positively correlate with the transcriptional activity of multiple genetic pathways. As proof, miRNA expression was associated with tumor size and texture enhancement [28]. Wu et al. found that BC tumor volume was positively correlated with the level of tumor-infiltrating lymphocytes (TILs), and the Cluster shade of the signal enhancement ratio was negatively correlated with TILs [29].

This paper discussed the differences in BC between various races from a radiogenomics perspective. First, we analyzed the diversity in imaging and omics expression between Chinese and Western BC derived from The Cancer Imaging Archive (TCIA) [30] and The Cancer Genome Atlas (TCGA) [31]. Then, imaging markers that can reflect gene expression activity were screened by establishing a mapping relationship between MRI image quantitative features and gene expression. Additionally, machine learning methods were used to verify the validity of these features. Finally, we analyzed the hub gene to explore its relationship to image features and clinical outcomes. The diagram of the whole scheme is shown in Figure 1.

## 2. Materials and Methods

### 2.1. Genomic and Picture Datasets

In this work, our main concern was to analyze the differences in image characteristics and transcriptome data between Chinese and Western BC patients. We derived the radiogenomic characterization using the below datasets:
Downloaded the TCGA-BRCA dataset from the TCIA database. To reduce the image quality difference between cases in multiple institutions, we selected MRI images obtained by the same scanner, and a total of 91 patients were obtained;Downloaded the gene expression RNAseq data from the GDC TCGA Breast Cancer dataset from UCSC Xena [32] (http://xena.ucsc.edu/ accessed on 1 November 2021). These transcriptome data correspond to patients with imaging data;Downloaded GSE116180 [33], GSE197894 [34], and GSE198545 [35] from the GEO database as validation datasets.


Currently, the public dataset lacks a Chinese breast cancer dataset containing imaging and omics data. We screened the corresponding patients in the TCGA-BRCA dataset according to the omics characteristics of Chinese BC patients to approximate them. It is known that the somatic mutagenic genes of Chinese breast cancer patients are PIK3CA, PIK3R1, AKT3, and PTEN [18], so we screened patients with the above mutations and labeled them as Chinese patients, and the remaining cases were Western patients.

### 2.2. Picture Data Analysis

TCIA imaging data were downloaded, and the image of the GE 1.5T instrument obtained was chosen. Image enhancement was performed by applying square, exponential, and wavelet transfers to the original image.

For the same patient, three different doctors marked the tumor area. According to the position marked by the doctor. Image features were obtained using the Python package ‘pyradiomics’ [36]. The extracted feature types included shape, first-order statistical, grayscale co-occurrence matrix, grayscale dependence matrix, grayscale run length matrix, gray level size zone matrix, and neighboring gray-tone difference matrix features. A total of 1033 image features were obtained for further research. The least absolute shrinkage and selection operator (Lasso) model was used for feature selection to avoid data redundancy. The filtered features were input into the RF model, and the AUC of classification was used to evaluate the features.

Through intra-group correlation analysis, the features were similar when extracted from the areas of interest marked by three different doctors. By taking the average value, other characteristics of each patient have unique values for subsequent analysis. 

### 2.3. Transcriptomedata Analysis

For the downloaded transcriptome data, we screened genes with encoded proteins, removed duplicate data by averaging, and selected genes with expression greater than 1 to remove low counts data. The R package ‘DESeq2’ [37] was used to identify DEGs between Chinese and Western BC patients, defining|log2 (fold change)|>1 and padj < 0.05. The volcanic map of DEGs was drawn with the R package ‘ggplot2’ [38]. After using the ‘org.hs.eg.db’ program to convert the DEGs identifier, the Gene Ontology (GO) [39] enrichment analysis of DEGs was carried out with the R package ‘clusterProfiler’ [40].

### 2.4. Association between Transcriptome and Image Features

In this step, we calculated the Pearson’s correlation coefficient between each image feature and the level of DEGs. For subsequent analysis, we only retained the significant correlation between image features and genes (*p* < 0.05). On this basis, genes with corr > 0.4 were screened for protein interaction network analysis (PPI) in string online databases [41], and the hub genes were screened in Cytoscape using cytoHubba [42] and MCODE [43]. We also investigated the correlation coefficients between image features and genes that indicate the linear relationship’s strength and direction.

### 2.5. Model Development and Statistical Analysis

In this experiment, RF models were developed using Python to evaluate features. The model parameters were optimized through the gridsearchcv function and 10-fold cross-validation, and the best parameters were selected in the learning process, such as criterion = ‘entry’, n_ Estimators = 50, etc. We randomly selected 30% of the data as the test set and 70% as the training dataset and added the stratify function to make the class distribution of the training and test set similar to that of the whole dataset.

We fed the image data, the DEGs, and the combination of these two types of data to the RF model separately. The AUC values were used to judge the classification performance of the three datasets. For the validation of the hub gene, ROC curve analysis was carried out through the R package ‘proc’ [44] to evaluate the performance of the hub gene in other datasets.

## 3. Results

### 3.1. Radiomic Features

The Lasso model was established to filter image features. We found λ.min and λ.1se through 20× cross-validation and built the model_lasso_min and model_lasso_1se model, respectively. We used a boxplot to visualize the predictions of both models and a Wilcoxon Signed-Rank Test to test whether the predictions were valid. Relative to model_lasso_1se, model_lasso_min performed better in terms of AUC values, as shown in Figure 2. So, we selected model_lasso_min to screen the image features and filtered out 47 image features (Table 1). 

### 3.2. Transcriptome Data Characteristics

There were remarkable differences in the transcriptome expression of patients in the two groups. A total of 328 DEGs were obtained, including 270 downregulated genes and 58 upregulated genes. Go enrichment analysis showed the potential biological functions of DEGs in BC. The downregulated genes are mainly involved in biological processes such as potential regulation, amine transport, and hormone secretion; the upregulated genes are primarily interested in sleep, molecular transmembrane transporter activity, and other functions, as shown in Figure 3.

### 3.3. Association between DEGs and Radiomics Features

#### 3.3.1. Diagnostic Role of Radiogenomics Signature

The radiogenomics correlation plot describes the Pearson’s correlation coefficient analysis between imaging features and DEGs (*p* < 0.05). Radiogenomics features comprised genetic data and imaging features that correlate greater than 0.4. We entered parts into the RF model and achieved an AUC of 0.95 in the validation dataset. The AUC value for the input of 47 imaging features was 0.74. The results suggest that, compared with imaging features or genetic data alone, radiogenomics features for the classification of Chinese and Western BC patients can further improve the classification validity of the model (Figure 4).

#### 3.3.2. Validation of the Hub Genes for the Differential Diagnosis of Chinese and Western BC Samples

We combined the results of cytoHubba and MCODE to get a total of 13 hub genes, as illustrated in Figure 5. The ROC curve verification of the hub genes was performed to verify the differences between these genes in Chinese and Western BC patients. We downloaded RNA sequence data from the GEO database for BC patients in China and the United States. A total of three datasets were downloaded, namely 12 cases of Chinese data in GSE116180, 10 cases of Chinese data in GSE197894, and 38 cases of American data in GSE198545, resulting in an extensive dataset of 22 Chinese patients and 38 cases of American patients.

As shown in Table 2, the AUC values of SYT4, GABRG2, CHGA, SLC6A17, NEUROG2, COL2A1, and MATN4 genes were more significant than 0.6, and the AUC values of GABRG2 and NEUROG2 reached 0.865 and 0.876, respectively. The values demonstrated substantial differences in these genes between the two groups of patients in the validation set, so the hub gene was finally determined as SYT4, GABRG2, CHGA, SLC6A17, NEUROG2, COL2A1, and MATN4.

#### 3.3.3. Association between Hub Genes and Imaging Features

Next, we studied the link between the hub genes and the imaging features. We found that the expression of the SYT4 and CHGA genes were positively associated with the square_firstorder_10Percentile imaging characteristics; the two genes were co-expressed in the protein-protein interaction network, and the regulatory pathway of catecholamine secretion was positively related to this feature; GABRG2 and COL2A1 were positively associated with square_ngtdm_Busyness, with the correlation value of GABRG2 reaching about 0.902, indicating that the γ-aminobutyric acid signaling pathway was related to the square_ngtdm_Busyness characteristics. SLC6A17 was involved in membrane potential regulation and was positively correlated with three image features, namely original_glszm_ZoneVariance, wavelet.HLL_glcm_JointEnergy, and wavelet. LHH_glcm_JointEnergy; NEUROG2 was negatively correlated with wavelet.HLL_firstorder_Median feature; and MATN4 was positively correlated with original_glszm_ZoneVariance (Figure 5).

## 4. Discussion

BC has noticeable ethnic differences [12], which are affected by factors such as environment, social development level, genetics [45], and lifestyle. Different gene mutations may lead to differences in drug resistance [46,47], and various gland and tissue characteristics may affect the type of surgery and the sensitivity to cancer detection [15]. To determine the mapping relationship between the radiogenomics characteristics, genes, and imaging characteristics of Chinese and Western BC patients, we comprehensively analyzed the combination of BC transcriptome and imaging data from TCGA and TCIA. Firstly, according to the doctor’s marked area of interest, the high-throughput image features were extracted by pyradiomics. The feature screening was realized by establishing the Lasso model, and 47 two-dimensional quantitative features were selected. Secondly, DESeq2 difference analysis was performed on genes, and GO enrichment was carried out to reveal their biological significance. The different genes and imaging characteristics were analyzed for the Pearson’s correlation coefficient; the highly relevant genes were input into the RF model, and the model’s performance was improved. Finally, we identified seven hub genes through cytoHubba, MCODE, and external data verification, further analyzing the relationship between hub genes and imaging features.

Through experiments, we found that the hub genes with different transcriptome data in Chinese and Western BC patients were SYT4, GABRG2, CHGA, SLC6A17, NEUROG2, COL2A1, and MATN. Through the verification of external data, the AUC values of these genes for the classification of patients in China and the United States were greater than 0.6. In addition, through the analysis of clinical data on TCGA-BRCA, we found that the SLC6A17 and NEUROG2 genes were related to the age of onset, and their expression was higher in the lower age group. CHGA is associated with survival in BC patients. Differences in the expression of MATN4 across stages of BC were statistically valid (Figure 6). Further analysis showed that the presentation of the hub gene would be shown in the image characteristics, and the distribution of voxel intensity, adjacent grayscale difference matrix, grayscale symbiosis matrix, gray level size area matrix, and other radiological features in the first-order feature image region were displayed.

For example, the expression of the SYT4 and CHGA genes is positively related to the square_firstorder_10Percentile. SYT4, which is mainly present in the Golgi body and cytosol of lymph nodes, belongs to the touch-binding protein (SYTs) family, which plays an essential part in the process of immune cells [48,49]. It is now known that SYT4 has a role in gastric adenocarcinoma and low-grade glioma and is associated with recurrence-free survival in BC [49]. CHGA encodes pheochromophilin A, or parathyroid secretory protein. It is a member of neuroendocrine secretory protein granules that reside in the secreting vesicles of neurons and endocrine cells, such as islets in the pancreas β secretory granules [48]. CHGA protein can be used as a potential biomarker for colon and breast neuroendocrine (NE) cancer diagnosis [50]. GABRG2 is mainly present in the cytoplasmic membrane, is involved in chemosynaptic transmission, and affects the expression of GABRA3, whose high expression level is inversely correlated with the survival rate of breast cancer patients, and which activates the Akt pathway and promotes the migration, invasion, and metastasis of breast cancer cells [51]. GABRG2 variants may be resistant to valproic acid [52], and our study found that GABRG2 is highly correlated with square_ngtdm_Busyness characteristics, and GABRG2 may also be a possible therapeutic target for breast cancer. COL2A1 was also positively related to square_ngtdm_Busyness. High COL2A1 expression delays the time to recurrence in high-grade plasmacytic ovarian cancer [53], and upregulation of COL2A1 reduces the migration and invasion of breast cancer cells [54]. The role of MATN4, SLC6A17, and NEUROG2 genes in breast cancer is currently unknown, but they play a role in other cancers, which may be an inspiration for future breast cancer gene research.

Linking imaging features to omics features is an evolving area of research that provides additional value for clinical imaging with relevant molecular biological information. One of our findings helps to study the differences in BC between different ethnic groups and implement precision medicine for the characteristics of BC patients in China. Limitations of our research include incomplete BC imaging genomics data; in the currently published dataset with both imaging and genetic data for a limited number of patients, we cannot fully assess the characteristic differences in imaging genetics between Chinese and Western BC patients, so this study downloaded sequencing data from Chinese and American BC patients from the GEO dataset for verification. In addition, we have to acknowledge that factors such as age, stage, and molecular subtype can cause differences between breast cancer patients. We found that these factors had a similar distribution between the two groups in our dataset (Fisher’s exact test, *p* > 0.05). Therefore, we mainly focused on the effect of the race on the results. As fundamental research work, we did not do clinical trials on related genes. Our work showed that these genes with protein expression not only have ethnic differences in expression but also cause differences in image characteristics, which may provide target genes for the precise treatment of breast cancer.

## 5. Conclusions

In conclusion, this study explored the differences in image and gene expression between Chinese and Western BC patients. Our results suggested that radiogenomics signatures are more differentiated between Chinese and Western patients than imaging and genes alone. We obtained hub genes of DEGs and found that the expression of these genes may be the factors that cause differences in age, survival, and stage between Chinese and Western BC patients. In addition, we found that the expression of these hub genes could be reflected in imaging features. Therefore, exploring the differences in radiogenomics between Chinese and Western BC patients helped understand the relationship between pathogenesis and imaging expression.

## Figures and Tables

**Figure 1 genes-13-02416-f001:**
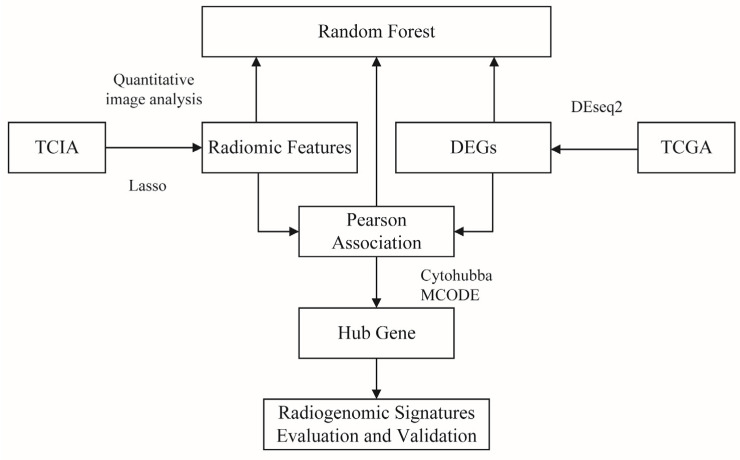
A generic flowchart of the proposed approach.

**Figure 2 genes-13-02416-f002:**
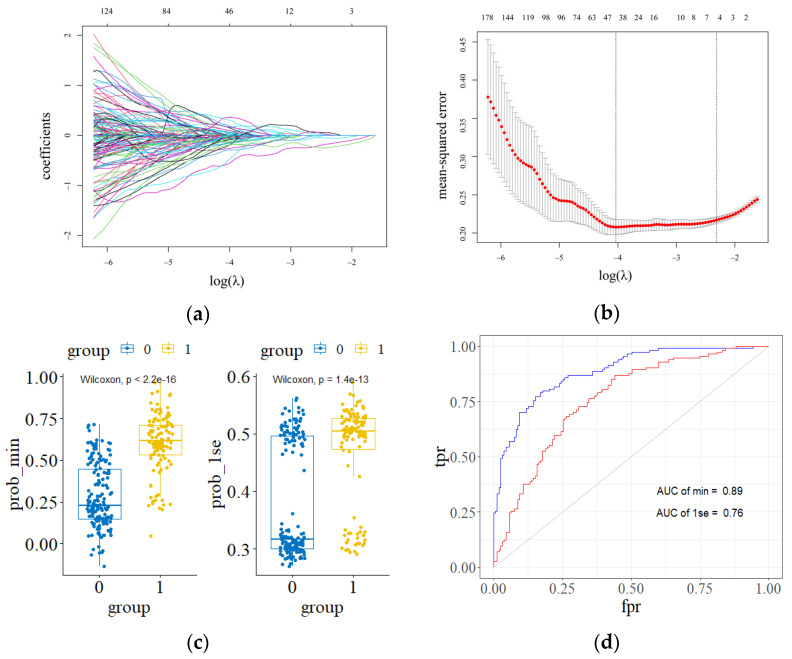
Lasso reduces the dimension of the feature. (**a**) Change track of the independent variable coefficient. (**b**) The dashed line on the left represents the λ.min value, and the dashed line on the right is λ.1se. (**c**) model_lasso_min and model_lasso_1se predictions. (**d**) The area under the ROC curve of the model_lasso_min and model_lasso_1se.

**Figure 3 genes-13-02416-f003:**
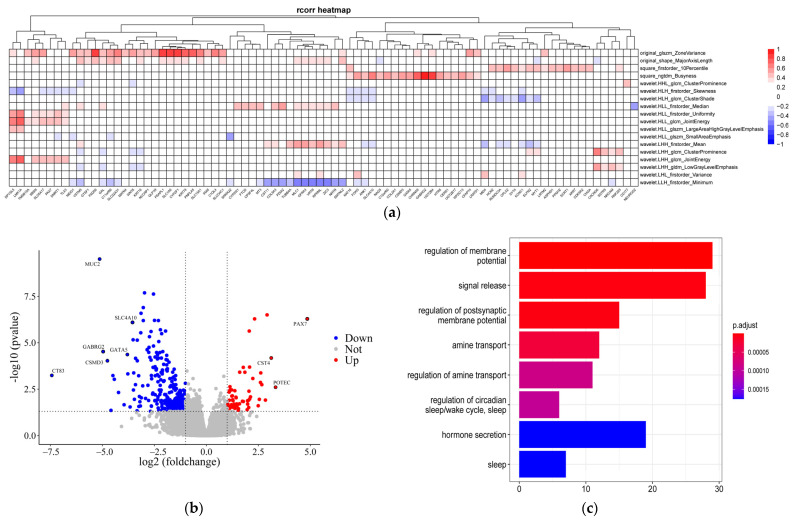
(**a**) Heatmap showing the association among imaging characteristics with genes (*p* < 0.05). Different colors represent correlation values, red squares indicate a positive correlation, and blue squares represent a negative correlation. (**b**) Differential gene volcano plot, blue points were the downregulated significant genes, and red points indicate upregulated significant genes. (**c**) GO enrichment analysis map.

**Figure 4 genes-13-02416-f004:**
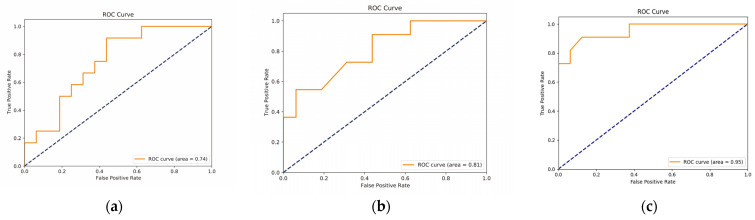
RF model results. (**a**) Result was obtained from 47 image feature input models. (**b**) Result of only differential genes input into RF. (**c**) Result obtained from radiogenomics features input models.

**Figure 5 genes-13-02416-f005:**
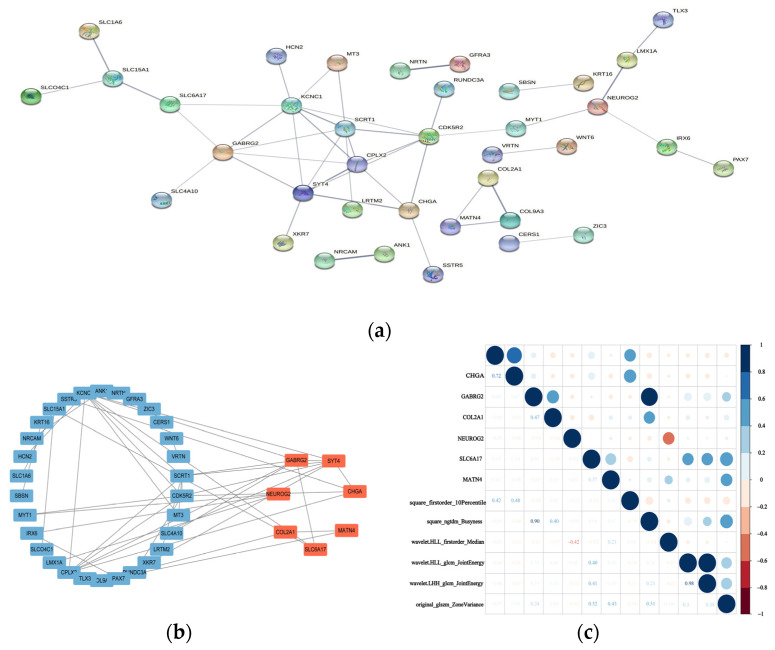
(**a**) Network diagram of the protein interactions of 84 genes. (**b**) The hub gene was validated by cytoHubba, MCODE, and external data. (**c**) Correlation coefficient diagram of hub gene and image characteristics (*p* < 0.05).

**Figure 6 genes-13-02416-f006:**
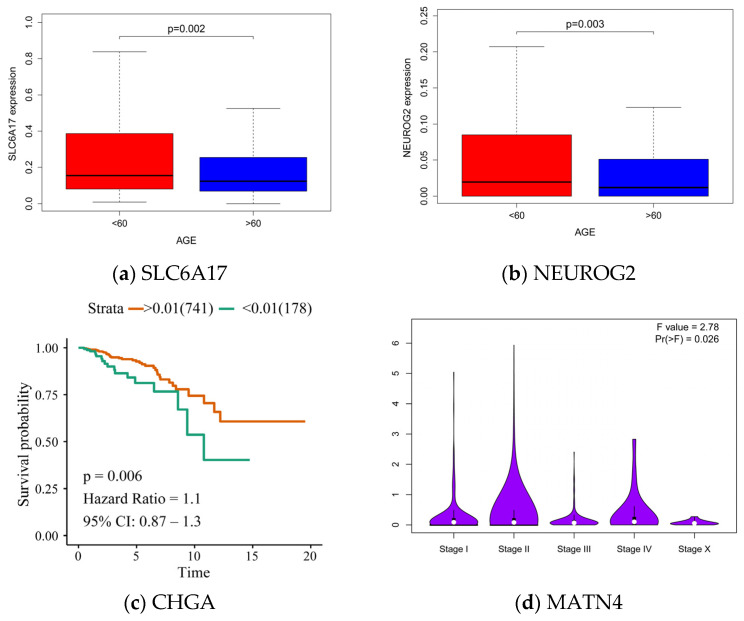
(**a**) The expression of the SLC6A17 gene between the two groups. (**b**) The expression of the NEUROG2 gene is relatively higher in the low age group. (**c**) The CHGA gene is related to the survival of BC patients. (**d**) The expression of MATN4 in different stages.

**Table 1 genes-13-02416-t001:** Forty-seven features were associated with RNA expression.

Type of Feature	Feature
Gray level co-occurrence matrix features (glcm)	Inverse Variance
	Correlation
	Informational Measure of Correlation
	Contrast
	Inverse Difference Moment Normalized
	Cluster Prominence
	Cluster Shade
	Joint Energy
	Maximal Correlation Coefficient
	Difference Variance
	Inverse Variance
Gray level size zone matrix features (glszm)	Small Area Emphasis
	Small Area, High Gray Level, Emphasis
	Zone Variance
	Zone Entropy
	High Gray Level Zone Emphasis
	Large Area, High Gray Level, Emphasis
	Gray Level, Non-Uniformity, Normalized
Shape features (2D)	Major Axis Length
	Maximum 2D diameter
First order features	Interquartile Range
	Robust Mean Absolute Deviation
	Tenth percentile
	Skewness
	Median
	Uniformity
	Mean
	Variance
	Minimum
Neighboring gray tone difference matrix features (ngtdm)	Busyness
	Complexity
	Coarseness
Gray level run length matrix features (glrlm)	Gray Level, Non-Uniformity, Normalized
	Low Gray Level Run Emphasis
	Gray Level Variance

**Table 2 genes-13-02416-t002:** The GEO dataset validated the results of the hub genes.

GENE	AUC
SYT4	0.628
GABRG2	0.865
CHGA	0.660
SLC6A17	0.605
NEUROG2	0.876
COL2A1	0.789
MATN4	0.693

## Data Availability

Publicly available datasets were analyzed in this study. This data can be found here: [https://www.cancerimagingarchive.net/ accessed on 1 November 2021].

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
