# Peer review of "Analysis of Breast Cancer Differences between China and Western Countries Based on Radiogenomics"

_genes, 2022, doi:10.3390/genes13122416_

Round 1

Reviewer 1 Report

Listed below are minor edits to the manuscript:

In section Materials and Methods 

Genomic and Imaging datasets:

1.     Please explain if the factors such as stage of tumor, age of the patient or subtype of the breast cancer considered while selecting the samples which could affect the outcome of the analysis?

2.     Were there any selection criteria applied to variants in genes PIK3CA, PIK3R1, AKT3, and PTEN such as VAF (variant allele frequency). Also please explain if one of the above or all the genes were a requirement for a sample to be considered to analysis.

In section Transcriptomics Data analysis

1.     Please elaborate on how the cleaning the downloaded transcriptome data was performed. Were factors such as biotype selection (only protein coding features selected or lincRNA and others), normalization factors, removing features not present in ‘x’ number of samples etc. considered while filtering the data? 

2.     Why was the adjusted p value not considered in identifying the significant genes? It could be a more robust filter to reduce noisy genes.

Figures:

1.     Figure1: Spelling error: “Evaluation”

2.     Figure3(b): It would be good to label the most significant genes (blue and red) on the differential gene volcano plots.

Line 93: It would be good to add a citation for study GSE197894.

Line87, 90,93: Please edit the word download to downloaded (to past tense) since the flow of text indicates the research was carried in the past.

Author Response

Dear Reviewers:

We are very grateful for your professional comments and careful examination on our article, and we apologize for our carelessness. Based on your comments, we have made textual corrections, presented the volcano map with important genes, and added a citation for the GSE197894 dataset in the resubmitted manuscript. We believe that your advice is of important scientific value and the explanation  about the questions in your comments are as follows:

Point 1: Please explain if the factors such as stage of tumor, age of the patient or subtype of the breast cancer considered while selecting the samples which could affect the outcome of the analysis?

Response 1: Your considerations make a lot of sense, we agree that the age, staging, and staging factors may have an impact on the results, as some results of the related work have shown. However, there are a small amount of data in the dataset, which may not produce representative results if the dataset is broken down into subgroup. And, after a test of the difference between the distribution of the age, stage, and subtype, we found that these factors had similar distribution between the two groups in our dataset (T-test or Fisher test, p>0.05). Therefore, we focused on the effect of the race on the results. For a more better results, the influence of age and staging has be taken into account in our study, and the work is undergoing. We have added this section to the manuscript limitations discussion.

Point 2: Were there any selection criteria applied to variants in genes PIK3CA, PIK3R1, AKT3, and PTEN such as VAF (variant allele frequency). Also please explain if one of the above or all the genes were a requirement for a sample to be considered to analysis.

Response 2:The main reason for our selection of these mutations is that the somatic mutations in PIK3CA (44%), PIK3R1 (17%), AKT3 (15%), and PTEN (12%) were stated to be prevalent in Chinese breast cancer patients, as shown in the work by LiChen.  Their mutation rates were all relatively high compared with TCGA data.  Respectively the PIK3CA and PIK3R1 mutation rates, at each stage are higher than that of the TCGA data. To investigate the relationship between genes and images, we chose the dataset TCGA-BRCA, which has the drawback of lacking Chinese patient data, and we grouped them by somatic mutation-prone genes. LiChen's article did not mention criteria such as VAF, so we analogized patients with mutations to Chinese breast cancer patients for analysis. We did hub gene validation on the GEO database with clear genetic data of Chinese and Western patients, and found that the hub genes, obtained in our work, also have Chinese and Western expression differences on other datasets, and our results were generalizable.

Chen, L.; Yang, L.; Yao, L.; Kuang, X.Y.; Zuo, W.J.; Li, S.; Qiao, F.; Liu, Y.R.; Cao, Z.G.; Zhou, S.L.; et al. Characterization of PIK3CA and PIK3R1 somatic mutations in Chinese breast cancer patients. Nat Commun 2018, 9, 1357, doi:10.1038/s41467-018-03867-9.

Point 3:how the cleaning the downloaded transcriptome data was performed? Were factors such as biotype selection (only protein coding features selected or lincRNA and others), normalization factors, removing features not present in ‘x’ number of samples etc. considered while filtering the data

Response 3: We downloaded counts data from UCSC Xena for differential analysis, which is in the form of log2(counts+1), so we first did an exponential transformation of it to obtain the raw counts data, selected genes encoding proteins, removed duplicate probe data with average de-duplication, and removed counts values less than or equal to 1 in all samples and removed genes containing missing values.

The FPKM data were downloaded from UCSC Xena for the correlation analysis, and the same data cleaning process was performed. The final FPKM data of the genes were obtained for subsequent analysis.

Thanks to your advice, we have added the process of data cleaning to the manuscript.

Point 4: Why was the adjusted p value not considered in identifying the significant genes? It could be a more robust filter to reduce noisy genes.

Response 4: Thank you for your advice, as you said the adjusted p-value is a more robust filter to reduce noisy genes, we used the adjusted p-value when selecting differential genes, but it was misspelled as p-value in the original article and has been changed in the original article.

We would like to thank the reviewers again for taking the time to review our manuscript.

Reviewer 2 Report

Dear Editor,

Thank you for the opportunity to review the manuscript Analysis of breast cancer differences between China and Western countries based on radiogenomics. This study aimed  to explore the differences in image and gene expression between Chinese and Western breast cancer patients. Authors suggest that radiogenomics signatures are more differentiated between Chinese and Western patients than imaging and genes alone. Authors obtained hub genes of DEGs and found that the expression of these genes may be the factors that cause differences in age, survival, and stage between Chinese and Western BC patients. In conclusion, exploring the differences in radiogenomics between Chinese and Western BC patients can help to understand the relationship between pathogenesis and imaging expression.  These findings could make the differences  between different ethnic groups and implement precision medicine for the characteristics of BC patients in China. These findings are very interesting and might provide a new perspective that will enable us to find potential novel targets for BC patients in China.

The weakness of the study is that research includes incomplete breast cancer imaging genomics data but the authors are aware of that. However manuscript is interesting and well-written and it is my opinion that can be considered for publication.

Author Response

Dear Reviewers:

Thank you for reading my manuscript in the midst of your busy schedule, and we are honored to receive your approval. We have refined some unsuitable expressions of the manuscript as much as possible, and we insure that these modifications do not affect the meaning of the paper. We are very grateful for your comments on our paper, which are a great encouragement to us.

Thank you and best wishes.

Reviewer 3 Report

Dear Author’s

I was pleased to review your intrereating paper.

I have the following comments:

- knowing this results do you think that women care can be improved?

-please explain the novelty of your study.

- minor english edits

- please explain the study limitation.

Author Response

Dear Reviewer:

Thank you for reading my manuscript despite your busy schedule, and thank you very much for your professional comments on our manuscript. The questions you raised are very meaningful for the improvement of our manuscript level, and for your queries, we make the following answers:

Point 1: knowing this results do you think that women care can be improved?

Response 1: Your considerations are valid. The seven hub genes not only had differential gene expression but also correlated with imaging features, which we obtained in Chinese and Western breast cancer patients. Some of the papers suggest that these genes may be clinically associated with breast cancer. We have added a discussion of hub genes in the manuscript that may become a target for future precision breast cancer therapy. SYT4 is associated with recurrence-free survival in breast cancer[1]; CHGA protein is a potential biomarker for the diagnosis of breast neuroendocrine (NE) cancer[2]; NEUROG2 is involved in the neural stem cell differentiation pathway and can regulate the regulation of luteinizing hormone biosynthesis and is currently found to be abnormally expressed in cervical cancer. GABRG2 is mainly present in the cytoplasmic membrane and is involved in chemosynaptic transmission, and affects the expression of GABRA3, the high expression level of which is inversely correlated with the survival rate of breast cancer patients, GABRA3 activates the Akt pathway and promotes the migration, invasion, and metastasis of breast cancer cells[3,4], therefore GABRG2 may also be a therapeutic target for breast cancer, and in the treatment of epilepsy. Additionally, the heterozygous variant of GABRG2 may be resistant to valproic acid[4], and it has also been suggested that GABRG2 may be associated with laryngeal cancer recurrence and colon tumors; the MATN4 gene encodes the Martilin4 protein, which is involved in cardiac remodeling and hematopoietic cell proliferation and may affect renal clear cell carcinoma progression and prognosis; high expression of COL2A1 delays high-grade plasmacytosis ovarian cancer and upregulation of COL2A1 reduces migration and invasion of MCF-7 and MDA-MB-231 breast cancer cells. miR-301 overexpression, which leads to low COL2A1 expression, can lead to increased migration and invasion and may be a potential therapeutic target[5,6]. SLC6A17 is mainly expressed in brain tissue and encodes a synaptic vesicular transporter of neutral amino acids and glutamate, which plays an important role in the regulation of glutamatergic synapses and is associated with mental retardation, and is not studied in cancer.

Point 2: please explain the novelty of your study.

Response 2: Breast cancer is highly heterogeneous, and the current diagnosis and treatment of breast cancer mainly rely on western data, which is not specific to Chinese breast cancer patients. Our study combines imaging and genes, which is a novel approach to more comprehensively analyze the heterogeneity between races, and our results show that differences in gene expression may lead to differences in imaging characteristics, suggesting that imaging characteristics may be able to be used in the future instead of genetic tests.

In addition, we compared the classification results of imaging and genetic data alone, and the classification accuracy was higher when combining the two factors under the same model, suggesting that the genomic features of imaging we obtained are more indicative of the differences between Chinese and Western breast cancer patients.

Point 3: minor english edits

Response 3: We are very grateful that you have reviewed this paper so carefully, and we apologize for the unsuitable expression in my manuscript. We worked on the manuscript for a long time and repeatedly added and removed sentences leading to poor readability. We have refined the wording and some unsuitable expression of the manuscript as much as possible, and we insure that these modifications do not affect the content and meaning of the paper.

Point 4: please explain the study limitation.

Response 4: Limitations of our study include incomplete breast cancer imaging genomics data, and we were unable to fully assess the differences in imaging genetic characteristics between Chinese and Western breast cancer patients due to the limited number of patients with both imaging and genetic data in the current publicly available dataset; therefore, this study downloaded sequencing data of Chinese and US breast cancer patients from the GEO dataset for validation of hub genes. In addition, we did not add the effect of age, stage, and subtype type in our study. This is because the distribution of these factors was similar between the two groups of patients in our data set (Fisher test, p>0.05). Therefore, we focused on the effect of race in our analysis. To obtain better results, the effects of age and staging have been taken into account in our study and this work is in progress. We have only conducted fundamental research work, lacking validation by drug analysis as well as clinical validation.

  1. Jiang, S.; Zhu, L.; Jiang, C.; Yu, S.; Wang, B.; Ren, Y. Prognosis and immune function of Synaptotagmin-4 in gastric cancer and brain low-grade glioma. 2020.
  2. Annaratone, L.; Medico, E.; Rangel, N.; Castellano, I.; Marchio, C.; Sapino, A.; Bussolati, G. Search for neuro-endocrine markers (chromogranin A, synaptophysin and VGF) in breast cancers. An integrated approach using immunohistochemistry and gene expression profiling. Endocr Pathol 2014, 25, 219-228, doi:10.1007/s12022-013-9277-4.
  3. Yan, L.; Gong, Y.Z.; Shao, M.N.; Ruan, G.T.; Xie, H.L.; Liao, X.W.; Wang, X.K.; Han, Q.F.; Zhou, X.; Zhu, L.C.; et al. Distinct diagnostic and prognostic values of gamma-aminobutyric acid type A receptor family genes in patients with colon adenocarcinoma. Oncol Lett 2020, 20, 275-291, doi:10.3892/ol.2020.11573.
  4. Shakir Ullah , N.A., Saad Ali , Adnan Khan , Sajjad Ahmad and Zia uddin. Distribution of Different Genotypes MTHFR and GABRG2 Genes in Epileptic Population of Khyber Pakhtunkhwa Pakistan. Clinical Schizophrenia & Related Psychoses 2020, doi:10.3371/CSRP.IASM.092320.
  5. Ganapathi, M.K.; Jones, W.D.; Sehouli, J.; Michener, C.M.; Braicu, I.E.; Norris, E.J.; Biscotti, C.V.; Vaziri, S.A.; Ganapathi, R.N. Expression profile of COL2A1 and the pseudogene SLC6A10P predicts tumor recurrence in high-grade serous ovarian cancer. Int J Cancer 2016, 138, 679-688, doi:10.1002/ijc.29815.
  6. Shi, W.; Gerster, K.; Alajez, N.M.; Tsang, J.; Waldron, L.; Pintilie, M.; Hui, A.B.; Sykes, J.; P'ng, C.; Miller, N.; et al. MicroRNA-301 mediates proliferation and invasion in human breast cancer. Cancer Res 2011, 71, 2926-2937, doi:10.1158/0008-5472.CAN-10-3369.
